# Antibody Response to Canine Parvovirus Vaccination in Dogs with Hyperadrenocorticism Treated with Trilostane

**DOI:** 10.3390/vaccines8030547

**Published:** 2020-09-19

**Authors:** Michèle Bergmann, Monika Freisl, Katrin Hartmann, Stephanie Speck, Uwe Truyen, Yury Zablotski, Matthias Mayr, Astrid Wehner

**Affiliations:** 1Clinic of Small Animal Medicine, LMU Munich, 80539 Munich, Germany; m.freisl@medizinische-kleintierklinik.de (M.F.); hartmann@lmu.de (K.H.); Y.Zablotski@med.vetmed.uni-muenchen.de (Y.Z.); m.mayr@medizinische-kleintierklinik.de (M.M.); a.wehner@medizinische-kleintierklinik.de (A.W.); 2Institute of Animal Hygiene and Veterinary Public Health, University of Leipzig, 04103 Leipzig, Germany; stephanie@speck-kaysser.de (S.S.); truyen@vetmed.uni-leipzig.de (U.T.)

**Keywords:** CPV, HAC, cushing, protection, immunosuppression, MLV, titer

## Abstract

It is unknown how dogs with hyperadrenocorticism (HAC) respond to vaccination. This study measured antibodies against canine parvovirus (CPV) in dogs with HAC treated with trilostane before and after CPV vaccination, and compared the immune response to that from healthy dogs. Eleven dogs with HAC, and healthy age-matched control dogs (*n* = 31) received a modified-live CPV vaccine. Antibodies were determined on days 0, 7, and 28 by hemagglutination inhibition. Univariate analysis was used to compare the immune response of dogs with HAC and healthy dogs. Pre-vaccination antibodies (≥10) were detected in 100% of dogs with HAC (11/11; 95% CI: 70.0–100) and in 93.5% of healthy dogs (29/31; 95% CI: 78.3–99.2). No ≥4-fold increase in antibody titer was observed in dogs with HAC while in 22.6% of healthy dogs, a ≥4-fold titer increase was observed (7/31; 95% CI: 11.1–40.1). Mild vaccine-associated adverse events (VAAEs) were detected in 54.5% of dogs with HAC (6/11; 95% CI: 28.0–78.8) and in 29.0% of healthy dogs (9/31; 95% CI: 15.9–46.8). There was neither a significant difference in presence of pre-vaccination antibodies (*p* = 1.000), or response to vaccination (*p* = 0.161), nor in the occurrence of VAAEs (*p* = 0.158). Immune function of dogs with HAC treated with trilostane seems comparable to that of healthy dogs.

## 1. Introduction

Canine parvovirus (CPV) is a highly infectious pathogen and still a major infectious cause of morbidity and mortality among the dog population worldwide [1]. Vaccination is strongly recommended for all dogs [2,3]. According to their ability to induce a stronger Th1-dominated immunity, modified live virus (MLV) vaccines are superior to inactivated vaccines in mediating protection against viral diseases; furthermore, they induce a longer duration of immunity [4]. Presence of serum antibodies against CPV correlates with protective immunity against CPV infection [5], and thus, measurement of antibodies can be used to evaluate the specific immune status of individual dogs.

It is currently unknown whether vaccination with MLV is safe and effective in dogs with hyperadrenocorticism (HAC). Safety is a concern as HAC can impair the dog’s immune system, and MLV vaccines might regain pathogenicity. Vaccine-induced immune stimulation might also increase susceptibility to secondary infections in patients with HAC [6].

Efficacy is a further concern of vaccination in dogs with HAC. Immune response to vaccination might not be comparable to the response in healthy dogs, and it is unclear, whether vaccines induce immunity at all and/or whether duration of immunity is shortened in dogs with HAC. In both human and veterinary medicine, there are no data on how individuals with HAC respond to vaccination with MLV. Thus, data in dogs are especially important since they can be used as model for vaccination in humans with HAC [7]. A previous study revealed that cats receiving glucocorticoid treatment were significantly less frequently protected against feline panleukopenia [8]. In contrast, a study in dogs demonstrated that a short-term treatment with different doses of glucocorticoids prior to or concurrently to vaccination did not significantly affect the immune response to MLV [9]. Nevertheless, international expert groups strongly recommend revaccination several weeks after systemic treatment with glucocorticoids to ensure development of immunity following vaccination [2,3]. 

Current guidelines suggest vaccinations in dogs with well-controlled HAC according to the proposed guidelines for healthy dogs [10]. However, similarly to long-term, systemic use of exogenous glucocorticoids or increased endogenous glucocorticoid levels in connection with stress [11], an inadequate immune response (mainly due to suppression of T-lymphocytes) must be expected [12] in dogs suffering from HAC especially if this condition is uncontrolled. In these cases, it is currently advised to postpone vaccinations until disease control is achieved [10].

The aim of this study was to determine differences regarding efficacy and safety of vaccination in dogs with HAC treated with trilostane compared to healthy dogs by measuring CPV antibodies within a period of 28 days after MLV vaccination.

## 2. Materials and Methods

### 2.1. Study Population 

The study was prospective. Dogs were included between November 2011 and April 2013. All dogs with HAC were patients of the Clinic of Small Animal Medicine, Centre for Clinical Veterinary Medicine, LMU Munich. All healthy dogs were presented to the Clinic of Small Animal Medicine, to a private practice in Southern Germany, or to a charity organization for vaccination. The study protocol was approved by the responsible veterinary authority, reference number 55.2-1-54-2532.3-61-11. 

Dogs were included if they had not received a CPV vaccine within the last 12 months. Dogs were excluded (1) if they had received any antibody preparations within the last 12 months or (2) if the required anamnestic data were not available (e.g., lack of current vaccination card).

Dogs from the HAC group had to have a documented history of a confirmed diagnosis of HAC, had to receive trilostane treatment at the time of vaccination, and to be clinically stable during the time of the study course. Diagnosis of HAC was based on history, physical examination findings and results of laboratory testing (hematology, biochemistry profile, and urinalysis) that are typically reported for dogs with HAC [13,14]. A further confirmation of diagnosis of HAC in all dogs was based on a lack of cortisol suppression (≥1.4 µg/dL) following low dose (0.01 mg/kg) dexamethasone administration or increased cortisol levels (>20 µg/dL) after adrenocorticotropic hormone (ACTH) stimulation. Additionally, dogs from the HAC group were examined for the presence of concurrent disorders or complications. The distinction between pituitary-dependent HAC (PDH) and a functional adrenal tumor (ADH) was based on ultrasonographic appearance of the adrenal glands and evidence of unilateral or bilateral adrenal gland enlargement [15] or results of the standard low-dose dexamethasone suppression test.

The healthy study group was age-matched (≥7 years of age) and included only dogs (1) without a history of illness, anesthesia, or surgery, (2) without a systemic drug treatment (except deworming) within the last 4 weeks, and (3) with an unremarkable physical examination. 

The dogs’ signalment, origin, environmental and housing conditions, vaccination history (previous vaccinations; complete vaccination series; time since last vaccination) as well as medical history were collected. 

### 2.2. Study Protocol

Health status of the dogs was examined by physical examination on days 0, 7, and 28. On day 0, dogs were vaccinated with a combined MLV vaccine containing CPV-2 strain 154 (viral titer of 10^7.0−8.4^ TCID_50_) as well as canine distemper virus (CDV), and canine adenovirus-2 (CAV-2) (Nobivac^®^ SHP, MSD); CDV and CAV-2 were not part of the present study. Owners were instructed to document possible vaccine-associated adverse events (VAAEs) until the end of the study. Serum samples were taken on days 0, 7, and 28 for measurement of antibodies before and after vaccination.

### 2.3. Detection of Antibodies by Hemagglutination Inhibition

Serum samples for CPV antibodies were frozen at −20 °C. Serum was analyzed by HI according to an established protocol by Carmichael et al., 1980 with minor modifications using 8 hemagglutinating units of CPV-2, strain vBI [16]. The highest dilution of serum that completely prevented hemagglutination inhibition (HI) of CPV antigen was considered as an endpoint. Plates were evaluated by 2 independent persons and divergent results were rechecked by a third independent person. All persons were blinded. 

Antibody titers ≥10 were regarded as positive (with 10 being the first dilution). Dogs with a 4-fold titer increase (2 titer steps) or higher were considered as “responding to vaccination” [17]. Dogs that had no CPV antibodies on day 0 and that did not develop antibodies during the study course were defined as “non-responders”.

### 2.4. Statistical Analysis 

Data were analyzed using R version 3.6.3 (2020-02-29) (R Foundation for Statistical Computing, Vienna, Austria). Shapiro wilk test was used to test numeric data for normality. Due to the non-normally distributed data, Wilcoxon test was used to determine significant differences between dogs with HAC and healthy dogs with regard to the dogs´ age, body weight, and time since last vaccination. Categorical data was analyzed with Fisher’s exact test to compare (1) presence of pre-vaccination antibodies between dogs with HAC and healthy dogs, (2) antibody response between dogs with HAC and healthy dogs, and (3) occurrence of VAAEs. A p-value below 0.05 was considered significant.

## 3. Results

### 3.1. Dog Population

The study included 11 dogs with HAC receiving medical treatment with trilostane (Table 1). Of these dogs, 7 dogs were female (63.6%) and 4 dogs were male (36.4%). Age ranged between 7 and 16 years (median age: 11 years). Body weight ranged between 7 and 46 kg (median weight: 18 kg). Dogs with HAC were mixed breed (*n* = 5; 45.5%) or purebred (*n* = 6; 54.5%). Ten dogs lived in an urban area (90.9%) and 1 dog in a rural area (9.1%). Eight dogs had 3 to 5 daily contacts with other dogs (72.7%), and 3 dogs (27.3%) had ≤2 daily contacts. PDH was diagnosed in 9 dogs (81.8%), and ADH was diagnosed in 2 dogs (18.2%). HAC was well-controlled in 6 dogs (54.5%), moderately controlled in 3 dogs (27.3%) and poorly controlled in 2 dogs (18.2%). Control of the disease was based on presence or resolution of clinical signs (e.g., polyuria, polydipsia and polyphagia), physical examination findings (e.g., haircoat, general appearance), laboratory data (e.g., urine concentration ability), and ACTH stimulation test results. The median dose of trilostane was 2.8 mg/kg/day (range, 0.8–13.2 mg/kg/day). Treatment with trilostane was started immediately after diagnosis of HAC in all dogs. A total of 9 dogs received trilostane once a day, 2 dogs received it twice a day. In one dog, concurrent diabetes mellitus was diagnosed when the dog entered the study. Median time between establishment of HAC diagnosis and vaccination on study day 0 was 270.6 days in the HAC group (range: 27–1185 days).

Of the 31 healthy dogs, 15 dogs were female (48.4%) and 16 dogs were male (51.6%). Age ranged between 7 and 13 years. Median age was 9 years. Body weight ranged between 3 and 43 kg. Median weight was 24 kg. Sixteen dogs were mixed breed (51.6%) and 15 dogs were purebred (48.4%). Eighteen dogs lived in an urban area (58.1%) and 13 dogs in a rural area (41.9%). Nineteen dogs had 3 to 5 daily contacts with other dogs (61.3%), and 12 dogs (38.7%) had ≤2 daily contacts. 

All dogs had received vaccinations in the past (more than one year ago). Two of the dogs from the HAC group (18.2%) and 2 of the healthy dogs (6.5%) had received a complete vaccination series according to current guidelines [2,3]. Median time since the last vaccination was 1.35 years (range: 1.01–7.13 years) in the HAC group and 1.71 years (range: 1.02–12.09 years) in the healthy group. A complete vaccination status was classified as a completed primary CPV vaccination series with a MLV in 3–4 week intervals and the last vaccination with at least 14–16 weeks, followed by a booster vaccination given 11–13 months later. In dogs older than 12 weeks, vaccination was considered complete, if they had received 2 vaccinations every 3–4 weeks followed by a booster after 11–13 months. After the primary vaccination series, dogs had to have received subsequent revaccinations in at least 3-year intervals [2,3]. 

Median age of the dogs from the HAC group (11 years) and the healthy dogs (9 years) differed significantly (*p* = 0.010). There was no significant difference between the median body weight (*p* = 0.383) and the median time since last vaccination (*p* = 0.597).

### 3.2. Response to Vaccination 

Table 2 and Table 3 summarize the response to vaccination of all dogs. Of the dogs with HAC, 100% (11/11; 95% CI: 62.8–100) had pre-vaccination antibodies ≥ 10 on day 0 (median antibody titer: 80; range: 20–160) (Table 1). None of these dogs had a ≥4-fold increase after vaccination (median antibody titer on day 7: 160 and on day 28: 160; range on day 7 and 28: each 20–320). VAAEs were observed in 54.5% of the dogs with HAC (6/11; 95% CI: 28.0–78.8) by the owners and included mild gastrointestinal symptoms (*n* = 4) or a slightly reduced general condition with less activity (*n* = 3) after vaccination for a few days; one dog showed both, mild gastrointestinal symptoms and a reduced general condition after vaccination.

Pre-vaccination antibodies ≥ 10 were present in 93.5% of the healthy dogs (29/31; 95% CI: 78.3–99.2) (median antibody titer: 160; range: 10–1280). Response to vaccination was observed in 22.6% (7/31; 95% CI: 11.1–40.1) of the healthy dogs (median antibody titer on day 7: 160 and on day 28: 160; range on day 7: 10–2560 and on day 28: 20–1280). No dog was identified as non-responder. In 29.0% (9/31; 95% CI: 15.9–46.8) of the healthy dogs, VAAEs were described by the owners and included mild gastrointestinal symptoms (*n* = 4) or a slightly reduced general condition with less activity (*n* = 5) after vaccination for a few days.

### 3.3. Comparison of Dogs with HAC and Healthy Dogs

There was no significant difference in the presence of pre-vaccination antibody titers ≥10 on day 0 (*p* = 1.000; odds-ratio (OR): 1.949; 95% CI: 0.087–43.82) between dogs with HAC and healthy dogs. Furthermore, there was no significant difference in the response to vaccination between dogs with HAC and healthy dogs (*p* = 0.161; OR: 7.041; 95% CI: 0.369–134.2). Despite a considerable difference in percentages, occurrence of VAAEs (*p* = 0.158; OR: 2.933; 95% CI: 0.710–12.11) was not significantly different in dogs with HAC compared to healthy dogs.

## 4. Discussion

HAC is a common disease in dogs caused by chronic elevation of blood cortisol. Thus, HAC can lead to immunosuppression [6], and it has been suggested that efficacy of vaccination might be reduced in immunosuppressed patients. However, data on the immune response in patients with HAC are missing in human as well as in veterinary medicine. 

In the present study, all dogs with HAC had pre-vaccination antibodies ≥10 on study day 0 suggesting that they had responded to vaccination or infection in the past, irrespective of the timing and stage of HAC. Given the fact that all dogs with HAC had pre-vaccination antibodies, they were likely protected against canine parvovirosis. The most important immunologic abnormalities leading to an impaired immune function in humans with HAC include a decrease in the number and proportion of CD4-positive lymphocytes that play an integral role in promoting and maintaining humoral and cell-mediated immunity [18]. However, in a well-controlled stage of disease, immunosuppression is likely mild, and a rapid and severe decrease of immune function only might occur in an uncontrolled stage [19]. In the present study, the majority of dogs with HAC were well or moderately controlled and, therefore, in a stage in which the immune function might be comparable to that of healthy dogs. Only two dogs were suffering from badly controlles HAC as clinical signs of hypercortisolism were still present. Therefore, those 2 patients could be considered as being more immunocompromised compared to the others, although both dogs had pre-vaccination antibodies. HAC in untreated dogs can lead to opportunistic bacterial infections—often present as pyoderma and/or urinary tract infections [20,21]. Those infections usually do not reoccur once HAC is controlled. HAC is biochemically characterized by non-physiological hypercortisolism that, similar to the use of exogenous glucocorticoids induces a reversible state of immunosuppression. The exact dose and duration of glucocorticoid excess that leads to immunosuppressive effects, however, is not well defined. A few studies evaluated the effect of exogenous glucocorticoid treatment on dogs´ immune system. Dogs treated with prednisolone (2 mg/kg every 12 h) showed a decrease in the serum concentration of all immunoglobulin classes as well as lower numbers of CD4- and CD8-positive T cells and B lymphocytes [22]. One study demonstrated a significant decrease in T cells after short-term use of prednisolone (3 days with a dose between 1.66 and 2.24 mg/kg every 24 h) over a period of 38 days [23]. However, since the severity of HAC as well as success of medical treatment varies between affected individuals, it might be difficult to compare certain glucocorticoid doses with the degree of spontaneous hypercortisolism. In addition, individual glucocorticoid sensitivity exists in humans and refers to the intensity at which a glucocorticoid-responsive system responds to glucocorticoids. It is hypothesized that tissue and target genes respond differently to glucocorticoids in every individual [24,25,26]. This variable sensitivity to cortisol likely also exists in dogs.

A few studies in dogs investigated the effect of glucocorticoid treatment on vaccination response. One study evaluated the effect of oral prednisolone on vaccination against canine distemper virus in Beagle puppies and found that doses of 1 mg/kg and 10 mg/kg prednisolone orally over a period of 21 days did not negatively affect response to vaccination [9]. The use of 0.25 mg/kg dexamethasone (which corresponds to a dose of 1.25 mg/kg prednisolone) in dogs before and after the first vaccination against rabies also had no negative effects on the antibody response [27]. Although immunocompromising effects of glucocorticoid treatment vary, a dose of 2 mg/kg prednisolone is considered as sufficiently immunocompromising to raise concerns about the safe use of MLV [28,29,30]. For example, dogs with experimental CPV infection developed encephalomyelitis after vaccination with older MLV vaccines against distemper, which was most likely favored by the immunosuppressive effect of concurrent CPV infection [30]. Something similar was shown after the application of older rabies MLV, which were available on the American market in the past. There are documented cases of vaccine-induced rabies in 11 dogs, 2 cats, and 1 fox after treatment with glucocorticoids [31]. Glucocorticoids used in lower (but still higher than physiologic) doses also might reduce the immune response to vaccines. In human medicine, glucocorticoid therapy usually does not contraindicate administration of MLV vaccines, when glucocorticoid therapy is short-term (less than 2 weeks) or given in only a low to moderate dose, or is administered only locally [32]. However, as stated before, external application of glucocorticoids is not necessarily comparable to an increased endogenous production.

The present study represents the first study evaluating the response to CPV vaccination of dogs with HAC. Data are important due to potential differences in response to vaccination depending on the individual immune status and they could serve as a model for vaccination in humans with HAC. Correct response to vaccination (≥4-fold titer increase) was observed in none of the dogs with HAC; 6 dogs had a titer increase but less than 2 titer steps that might be considered to be within experimental error. However, also the majority of healthy dogs (77.4%) did not respond to vaccination. The most obvious explanation for the lack of antibody increase in both groups is the presence of pre-vaccination antibodies. A lack of response to modified live CPV vaccination in dogs with pre-vaccination CPV antibodies (titer ≥ 80) already has been demonstrated in healthy dogs [33] and might be explained by the binding of the pre-existing antibodies to the vaccine virus. Therefore, CPV routine re-vaccination is not recommended in adult dogs with pre-existing antibodies. 

Some dogs with HAC had concurrent conditions (Table 1). One dog from the HAC group in the present study was presented with newly diagnosed diabetes mellitus. Diabetes mellitus can lead to deviations in macrophage function [34]. Various studies in human medicine have shown that vaccinations during therapy with insulin or oral antidiabetic drugs are effective and safe [35,36,37], although, in poorly controlled patients, hyperglycemia is assumed to strongly compromise the immune system. The immune response of the diabetic dog in the present study, however, was comparable to that of the other dogs and the dog showed no VAAEs.

It has to be considered that immunity in dogs with HAC was not markedly compromised since all dogs had been vaccinated in the past prior to development of HAC. Humoral immune response to de novo vaccination in dogs with already existing HAC might however be impaired due to a lower number of naive T cells [11,38]. 

So far, there are no data on whether dogs with HAC might be at increased risk for developing VAAEs after vaccination with MLV. Gastrointestinal symptoms and lethargy after vaccination were commonly observed in the present study, presumably because owners were advised to pay special attention to occurrence of VAAEs. Although there was no significant difference in occurrence of VAAEs, more than half of the dogs in the HAC group showed VAAEs. Glucocorticoid excess has several effects on the immune system e.g., deviations in neutrophil and macrophage function, such as impaired chemotaxis, phagocytosis, and bactericidal activity as well as decreased interleukin-1 production and antigen-processing and can predispose dogs with HAC to secondary infections [6,11,20,21]. Since clinical signs were mild and transient, they were likely a direct result of active virus replication in the lymphoid tissue and epithelial cells of the gastrointestinal tract. Replication of MLVs might be increased in dogs with HAC due to a declined function of the primary immune system (e.g., natural killer cells) being a first-line defense mechanism against viral infections [39]. Results indicate that vaccination in dogs with HAC is as safe as in healthy dogs at least in the short term. However, it is possible that the present study failed to demonstrate a statistically significant difference in occurrence of VAAEs between dogs with HAC and healthy dogs due to the low number of dogs with HAC. Rebound immunity after HAC remission is a concern in humans, frequently resulting in overt immune-mediated disease, either shortly after HAC treatment or delayed for several months. Until then, immune-mediated diseases remain unrecognized likely due to the high levels of cortisol that prevent humans with HAC from the development of clinical signs. The spectrum includes thyroid autoimmune disease and allergic diseases [40]. Similar to humans, rebound immunity in dogs treated for HAC is possible but data are missing. Nevertheless, vaccination has been discussed to be one factor that could contribute to the development of canine immune-mediated diseases. Overstimulation of the immune system might lead to activation of T cells and self-reactive B cells, causing autoantibody production typically against erythrocytes and/or thrombocytes but also against thyroglobulin [41,42,43]. Cells are destructed by cytolysis via the complement system, phagocytosis via macrophages or Fc receptor-mediated natural killer cells via antibody-dependent cell-mediated cytotoxicity. This explains why clinical consequences can occur at different time points; destruction via macrophages or natural killer cells can take days to weeks, whereas cytolysis via the complement systems is much faster [44]. Considering a possible combination of predisposing factors for development of immune-mediated diseases in dogs treated for HAC that are presented for regular re-vaccination, antibody testing should be emphasized particularly in these dogs and re-vaccination should only be considered in dogs without CPV antibodies. 

This study included age-matched adult control dogs (≥7 years). Younger dogs were not included in the study in order to rule out age influence on immune response. A lower number of circulating lymphocytes is known to cause a reduced immune response in elderly people [45]. Studies in dogs demonstrated an age-related remodeling of the immune system, with a gradual decline in relative percentage of lymphocytes in dogs older than 7 years compared to younger dogs [46]. 

The main limitation of the study was the relatively small number of dogs with HAC that makes a clear assessment of the response to vaccination difficult. Owners of dogs with HAC were very reluctant to vaccinate their dogs, even if they were well-controlled, due to fear of progression of HAC or VAAEs. Further long-term studies should be performed involving larger numbers of dogs with HAC. In addition, measuring CD4+ and CD8+ cells would be desirable to further investigate cellular immunity. 

## 5. Conclusions

All dogs with HAC had pre-vaccination antibodies against CPV, indicating protection. Vaccination response in dogs with HAC was similar to that of healthy dogs and, thus, immune function seems to be comparable. However, mild VAAEs were commonly observed after vaccination. Antibody measurement against CPV infection would therefore be an excellent possibility in dogs with HAC to confirm that protection is present instead of routine re-vaccination. Further long-term studies should be performed involving larger numbers of dogs with HAC and at different stages of disease. 

## Figures and Tables

**Table 1 vaccines-08-00547-t001:** Signalment, history, and pre-vaccination antibody titer against canine parvovirus (CPV) of the dogs with hyperadrenocorticism (HAC) treated with trilostane and their response to vaccination.

Dog	Signalment and Weight	Origin of HAC (Time Since Diagnosis in Days)	Time Since Treatment in Days (Current Trilostane Dose and Control of HAC)	Concurrent Condition (Treatment)	Origin	Daily Contact to Other Dogs, Environment	Time Since Last Vaccination	Previous Vaccinations	Complete Vaccination Series	CPV Antibody Titer			≥4 Fold Titer Increase after MLV ^1^ Vaccination	VAAEs ^2^
1	Dalmatian, 10 y ^3^, m ^4^ n ^5^, 38.5 kg	PDH ^6^ (1185)	1184 (1.7 mg/kg BID ^7^, good control)	hypertension (amlodipine)	private	3 to 5, urban	2 y	7, 8 months; 2, 3, 4, 5, 6, 7, 8 y	yes	80	80	80	no	none
2	Mix, 15 y, f ^8^ s ^9^, 26.5 kg	ADH ^10^ (27)	26 (1.9 mg/kg SID ^11^, good control)	none	animal welfare	3 to 5, urban	2 y	7, 9, 10, 11, 12, 13 y	no	20	20	40	no	mild gastrointestinal symptoms on days 10–13 after vaccination
3	Mix, 12 y, mn, 9.2 kg	PDH (321)	320 (1.1 mg/kg SID, good control)	hypertension (amlodipine)	animal shelter	3 to 5, urban	7 y	2, 4.5 y	no	160	320	320	no	none
4	Tibet Terrier, 10 y, fs, 12.4 kg	ADH (360)	359 (2.4 mg/kg SID, good control)	chronic bronchitis (none)	breeder	<2, urban	1 y	2, 3, 4, 5, 6, 9 y	no	80	160	160	no	mild reduced general condition on day 1 after vaccination
5	Mix, 17 y, mn, 8.9 kg	PDH (105)	104 (1.2 mg/kg SID, good control)	mitral valve disease (none), cataract (none)	animal shelter	3 to 5, urban	1 y	14, 15, 16 y	yes	40	40	80	no	mild reduced general condition on days 0–1 after vaccination
6	Mix, 12 y, fs, 17.5 kg	PDH (40)	39 (1.7 mg/kg SID, good control)	none	animal welfare	<2, urban	1 y	3, 4, 5, 6, 7, 9, 10.5 y	no	80	160	160	no	mild gastrointestinal symptoms on days 0–2 days after vaccination
7	Beagle, 9 y, fs, 18.6 kg	PDH (364)	363 (6.6 mg/kg BID, moderate control)	none	animal welfare	3 to 5, urban	1 y	6 weeks; 2, 3 months, 1, 1.5, 5, 8 y	no	80	80	160	no	mild gastrointestinal symptoms on days 2–3 days after vaccination
8	Hovawart, 10 y, fn, 46 kg	PDH (91)	90 (1.3 mg/kg SID, moderate control)	urinary incontinence (caninephrin)	breeder	<2, rural	1 y	3, 4, 5, 6, 7, 8, 9 y	no	20	20	20	no	none
9	Magyar Viszla, 14 y, fs, 27 kg	PDH (78)	77 (2.2 mg/kg SID, moderate control)	urinary incontinence (phenylpro-panololamin)	breeder	3 to 5, urban	4.5 y	7, 9 weeks; 1, 2, 3, 5, 6, 7.5, 9 y	no	160	160	160	no	none
10	Yorkshire Terrier, 8 y, fs, 7.3 kg	PDH (309)	308 (1.5 mg/kg SID, poor control)	diabetes mellitus (Caninsulin^®^), tracheal collapse	breeder	3 to 5, urban	4 y	2, 3, 4, 12, 15 months; 4 y	no	160	160	160	no	none
11	Mix, 8 y, male, mn, 12.5 kg	PDH (97)	96 (0.8 mg/kg SID, poor control)	ventricular tachyarryth-mia (sotalol)	private	3 to 5, urban	1 y	2, 3, 15 months; 2, 3, 4, 5, 6, 7 y	no	320	320	320	no	mild reduced general condition and mild gastrointestinal symptoms on days 0–7 days after vaccination

^1^ MLV = modified life virus; ^2^ VAAEs = vaccine-associated adverse event; ^3^ y = years; ^4^ m = male; ^5^ n = neutered; ^6^ PDH = pituitary-dependent hyperadrenocorticism; ^7^ BID = twice daily; ^8^ f = female; ^9^ s = spayed; ^10^ ADH = adrenal-dependent hyperadrenocorticism; ^11^ SID = once daily.

**Table 2 vaccines-08-00547-t002:** Comparison of the humoral immune response of dogs with hyperadrenocorticism (HAC) treated with trilostane and healthy dogs after modified live virus vaccination against canine parvovirus using Fisher´s exact tests.

Total		Total	Dogs with HAC	Healthy Dogs	*p* ^1^
			11	31	
pre-vaccination antibody titer (*n* ^2^ = 42)	<10	2	0/11 (0.0%)	2/31 (6.5%)	1.000
	≥10	40	11/11 (100.0%)	29/31 (93.5%)	
≥4-fold titer increase (*n* = 42)	yes	7	0/11 (0.0%)	7/31 (22.6%)	0.161
	no	35	11/11 (100.0%)	24/31 (77.4%)	
vaccine-associated adverse events ^3^	yes	15	6/11 (54.5%)	9/31 (29.0%)	0.158
	no	27	5/11 (45.5%)	22/31 (71.0%)	

^1^*p* = *p*-value; ^2^
*n* = number of dogs; ^3^ based on owner reports and veterinary examination on days 7 and 28.

**Table 3 vaccines-08-00547-t003:** Canine parvovirus pre-vaccination antibody titer on day 0 and number of dogs with an ≥4-fold titer increase during the course of the study.

Pre-Vaccination CPV ^1^ Antibody Titer on Day 0	Total	Number of Dogs with a ≥4-Fold Antibody Titer Increase with the Respective Pre-Vaccination Antibody Titer on Day 0	
		Dogs with HAC ^2^	Healthy Dogs
<10	2	0/0 (0.0%)	2/2 (100.0%)
10	1	0/0 (0.0%)	1/1 (100.0%)
20	2	0/2 (0.0%)	0/0 (0.0%)
40	5	0/1 (0.0%)	0/4 (0.0%)
80	9	0/4 (0.0%)	1/5 (20.0%)
160	13	0/3 (0.0%)	2/10 (20.0%)
320	7	0/1 (0.0%)	1/6 (16.6%)
640	2	0/0 (0.0%)	0/2 (0.0%)
1280	1	0/0 (0.0%)	0/1 (0.0%)
Total number of dogs with ≥4-fold antibody titer increase		0/11 (0.0%)	7/31 (22.6%)

^1^ CPV = canine parvovirus, ^2^ HAC = hyperadrenocorticism.

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
