# Peer review of "Antibody Response to Canine Parvovirus Vaccination in Dogs with Hyperadrenocorticism Treated with Trilostane"

_vaccines, 2020, doi:10.3390/vaccines8030547_

Round 1
Reviewer 1 Report
Reviewer comments for manuscript ID vaccines-928271 entitled ‘Antibody response to canine parvovirus vaccination in dogs with hyperadrenocorticism treated with trilostane’
General Comments
An excellent study conducted meticulously on a clinically relevant topic of immense value to practising vets. Frankly, I was not able to point out much flaws in the research design except that sex of the recruited dogs was not taken into consideration in the analysis that I feel might have clinical relevance. This limitation should be mentioned in the conclusions. Discussion at few places goes off track. I suggest the authors to restrict discussion relevant to the hypotheses and results of the present study. Overall, I highly recommend this manuscript for publication after undergoing minor suggestions listed.
Abstract
Please modify the sentence ‘A ≥4-fold titer increase was observed in none of the dogs with HAC and in 22.6% of healthy dogs (7/31; CI95%: 11.1–40.1)’ as ‘No increase in antibody titre was observed in dogs with HAC while in 22.6% of healthy dogs a ≥4-fold titer increase was observed (7/31; CI95%: 11.1–40.1)’
Introduction
Please clarify this sentence ‘Nevertheless, international expert groups strongly recommend revaccination several weeks after systemic treatment with glucocorticoids had been discontinued to ensure development of immunity after vaccination [2, 3]’. I feel it should read as ‘Nevertheless, international expert groups strongly recommend revaccination several weeks after systemic treatment with glucocorticoids to ensure development of immunity after vaccination [2, 3]’
Materials and methods
Study population
I suggest, please change ‘Suspicion of HAC was based….’ To ‘Confirmation of HAC was based….’
Please modify ‘A diagnosis of HAC was confirmed in all dogs…..’ to ‘A further confirmation of diagnosis of HAC in all dogs was……’
Results
Table 2: Please rewrite n2 as 2n.
Author Response
Reviewer 1
Open Review
(x) I would not like to sign my review report
( ) I would like to sign my review report
English language and style
( ) Extensive editing of English language and style required
( ) Moderate English changes required
(x) English language and style are fine/minor spell check required
( ) I don't feel qualified to judge about the English language and style
|
Yes |
Can be improved |
Must be improved |
Not applicable |
|
|
Does the introduction provide sufficient background and include all relevant references? |
(x) |
( ) |
( ) |
( ) |
|
Is the research design appropriate? |
(x) |
( ) |
( ) |
( ) |
|
Are the methods adequately described? |
(x) |
( ) |
( ) |
( ) |
|
Are the results clearly presented? |
(x) |
( ) |
( ) |
( ) |
|
Are the conclusions supported by the results? |
(x) |
( ) |
( ) |
( ) |
Comments and Suggestions for Authors
Reviewer comments for manuscript ID vaccines-928271 entitled ‘Antibody response to canine parvovirus vaccination in dogs with hyperadrenocorticism treated with trilostane’
General Comments
An excellent study conducted meticulously on a clinically relevant topic of immense value to practising vets.
We thank the reviewer for this positive comment.
Frankly, I was not able to point out much flaws in the research design except that sex of the recruited dogs was not taken into consideration in the analysis that I feel might have clinical relevance. This limitation should be mentioned in the conclusions.
Sex was not taken into consideration in the analysis in the present study since several field studies failed to confirm sex-related associations with the immune function in dogs especially with regard to the prevalence of anti-CPV antibodies (Tennant et al., 1991; McCaw et al., 1998; Twark and Dodds, 2000; Almendra et al., 2005; Schoder et al., 2006; Corrain et al., 2007; Levy et al., 2008; Yang et al., 2010; Taguschi et al., 2011; Belsare et al., 2014; Castanheira et al., 2014; Orozco et al., 2014; Saasa et al., 2016; Kim et al., 2017; Kim et al., 2018), but also with regard to titer increase after vaccination i. e. after rabies vaccination (Mansfield et al., 2004; Kennedy et al., 2007). However, we now looked for significant differences between the healthy female and the healthy male dogs from the present study with regard to presence of anti-CPV antibodies and titer increase using Fisher´s exact test. There was no significant difference in the presence of pre-vaccination antibody titers ≥10 on day 0 (p = 1.000; odds-ratio (OR): 1; 95% CI: 0.012–83.978) between healthy female dogs and healthy male dogs. There was also no significant difference in titer increase (p = 1.000; odds-ratio (OR): 0.757; 95% CI: 0.091–5.581) between healthy female dogs and healthy male dogs. We leave it up to the reviewer and editor to include this analysis.
Discussion at few places goes off track. I suggest the authors to restrict discussion relevant to the hypotheses and results of the present study. Overall, I highly recommend this manuscript for publication after undergoing minor suggestions listed.
Since reviewer 2 appreciated all the details, only minor changes have been made (at the request of reviewer 2).
Abstract
Please modify the sentence ‘A ≥4-fold titer increase was observed in none of the dogs with HAC and in 22.6% of healthy dogs (7/31; CI95%: 11.1–40.1)’ as ‘No increase in antibody titre was observed in dogs with HAC while in 22.6% of healthy dogs a ≥4-fold titer increase was observed (7/31; CI95%: 11.1–40.1)’
We modified the sentence accordingly.
“No increase in antibody titer was observed in dogs with HAC while in 22.6% of healthy dogs, a ≥4-fold titer increase was observed (7/31; CI95%: 11.1–40.1).”
Introduction
Please clarify this sentence ‘Nevertheless, international expert groups strongly recommend revaccination several weeks after systemic treatment with glucocorticoids had been discontinued to ensure development of immunity after vaccination [2, 3]’. I feel it should read as ‘Nevertheless, international expert groups strongly recommend revaccination several weeks after systemic treatment with glucocorticoids to ensure development of immunity after vaccination [2, 3]’
Yes, we thank the reviewer for this input. We made the respective changes.
“Nevertheless, international expert groups strongly recommend revaccination several weeks after systemic treatment with glucocorticoids to ensure development of immunity following vaccination [2, 3].”
Materials and methods
Study population
I suggest, please change ‘Suspicion of HAC was based….’ To ‘Confirmation of HAC was based….’
Please modify ‘A diagnosis of HAC was confirmed in all dogs…..’ to ‘A further confirmation of diagnosis of HAC in all dogs was……’
We made the following changes.
“Diagnosis of HAC was based on history, physical examination findings and results of laboratory testing (hematology, biochemistry profile, and urinalysis) that are typically reported for dogs with HAC [13, 14]. A further confirmation of diagnosis of HAC in all dogs was based on a lack of cortisol suppression (≥1.4 µg/dL) following low dose (0.01 mg/kg) dexamethasone administration or increased cortisol levels (>20 µg/dL) after adrenocorticotropic hormone (ACTH) stimulation.”
Results
Table 2: Please rewrite n2 as 2n
We are not sure whether this is in line with the paper´s format. We leave it up to the editor to change this.
Reviewer 2 Report
The paper entitled “Antibody response to canine parvovirus vaccination in dogs with hyperadrenocorticism treated with trilostane” is very interesting and well designed. The topic is relevant considering that more and more frequently vets have the problems to evaluate the possibility of vaccination in ill animals.
I appreciated a lot the details and accuracy of the data and statistical analysis.
The paper is very well written and I have only some minor curiosities to ask.
In the study population, what is the time frame for the enrollment of dogs (e.g. one year between 2018-2019)? This could be useful for setting the temporal context of the analyzes over time.
Section 2.3: generally we consider 4 dilutions to calculate seroconversion; you have to evaluate a 4 fold titer increase (only 2 titer steps). Is it only for a question of titer because it is based on 1:10 or it is a general approach? We consider 4 dilutions, regardless the test, because the load of antigen/antibodies and their binding vary according to the test.
The discussion led me to think that perhaps also a simple blood test could, albeit in part, satisfy some questions about the white blood cell situation, to be used as an immunity proxy. I think these parameters were evaluated in these dogs due to the accuracy of the study and even a simple sentence summarizing the behaviour of these cells, reporting “data not shown”, could be useful.
Finally, the authors have talked extensively about cellular immunity and CD4 + evaluation in the discussion. In the last sentence of the discussions they could add that further studies should also analyze the behaviour of cellular immunity evaluating/counting immunity cells like CD4+ and others.
Author Response
Reviewer 2
Open Review
( ) I would not like to sign my review report
(x) I would like to sign my review report
English language and style
( ) Extensive editing of English language and style required
( ) Moderate English changes required
(x) English language and style are fine/minor spell check required
( ) I don't feel qualified to judge about the English language and style
|
Yes |
Can be improved |
Must be improved |
Not applicable |
|
|
Does the introduction provide sufficient background and include all relevant references? |
(x) |
( ) |
( ) |
( ) |
|
Is the research design appropriate? |
(x) |
( ) |
( ) |
( ) |
|
Are the methods adequately described? |
(x) |
( ) |
( ) |
( ) |
|
Are the results clearly presented? |
(x) |
( ) |
( ) |
( ) |
|
Are the conclusions supported by the results? |
(x) |
( ) |
( ) |
( ) |
Comments and Suggestions for Authors
The paper entitled “Antibody response to canine parvovirus vaccination in dogs with hyperadrenocorticism treated with trilostane” is very interesting and well designed. The topic is relevant considering that more and more frequently vets have the problems to evaluate the possibility of vaccination in ill animals.
I appreciated a lot the details and accuracy of the data and statistical analysis.
We thank the reviewer for the positive comments.
The paper is very well written and I have only some minor curiosities to ask.
In the study population, what is the time frame for the enrollment of dogs (e.g. one year between 2018-2019)? This could be useful for setting the temporal context of the analyzes over time.
We added the information.
The study was prospective. Dogs were included between November 2011 and April 2013.
Section 2.3: generally we consider 4 dilutions to calculate seroconversion; you have to evaluate a 4 fold titer increase (only 2 titer steps). Is it only for a question of titer because it is based on 1:10 or it is a general approach? We consider 4 dilutions, regardless the test, because the load of antigen/antibodies and their binding vary according to the test.
In our test protocol sera are generally diluted 1:5, giving the lowest antibody titer to be calculated as 10. We have always considered a variation of 1 dilution step as a natural variation which can be due to minor inaccuracies in virus or serum dilution or cell number seeded. Therefore, we consider 2 dilution steps as significant and indicative for a titer increase, and this was always confirmed by the values of duplicates or triplicates. Four dilution steps as a benchmarks appears exaggerated, as at our lowest dilution (1:10) this would give a minimum titer of 1:160 to be accepted as a "real" increase.
The discussion led me to think that perhaps also a simple blood test could, albeit in part, satisfy some questions about the white blood cell situation, to be used as an immunity proxy. I think these parameters were evaluated in these dogs due to the accuracy of the study and even a simple sentence summarizing the behaviour of these cells, reporting “data not shown”, could be useful.
We are unsure, if we understood correctly: lymphopenia could be present in dogs with untreated HAC. Successful disease control could alleviate this laboratory abnormality and indicate regain of immunity. However, in a retrospective analysis of 175 dogs with HAC in our hospital population, only 28% were lymphopenic and 10% had a differential white blood cell count compatible with a stress leukogram (lymphopenia and neutrophilia) (data unpublished). This would make the lymphocyte count an unsuitable marker to monitor immunity.
Unfortunately, hematology was not part of the study protocol and is therefore only available in exceptional cases (if medically indicated).
Finally, the authors have talked extensively about cellular immunity and CD4 + evaluation in the discussion. In the last sentence of the discussions they could add that further studies should also analyze the behaviour of cellular immunity evaluating/counting immunity cells like CD4+ and others.
We added this information.
In addition, measuring CD4+ and CD8+ cells would be desirable to further investigate cellular immunity.